



# MESMER-RCM: A Probabilistic Climate Emulator for Regional Warming Projections

Hao Pan[1], Lukas Gudmundsson[1], Mathias Hauser[1], Jonas Schwaab[2,3], Yann Quilcaille[1], and Sonia I. Seneviratne[1]

[1]Institute for Atmospheric and Climate Science, Department of Environmental Systems Science, ETH Zurich, Zurich, Switzerland
[2]WSL, Institute for Snow and Avalanche Research (SLF), Davos Dorf, Switzerland
[3]Institute for Environmental Planning, Leibniz Universität Hannover, Germany

**Correspondence:** Hao Pan (haopan8080@gmail.com)

**Abstract.** Regional Climate Model (RCM) emulators enable rapid and computationally efficient RCM projections given Global Climate Model (GCM) inputs, complementing dynamical downscaling by approximating physical representations with statistical models. However, while existing RCM emulators perform well in deterministic emulations, they do not sample internal RCM variability and remain computationally expensive. Here, we present MESMER-RCM, a probabilistic RCM emulator designed for spatially resolved annual 2m temperature. MESMER-RCM is a generative model that enables both data-efficient learning and interpretability. It can generate large ensembles of synthetic, yet physically plausible, RCM realizations, capturing the internal RCM variability at a fraction of the computational cost. This work offers a fast and reliable RCM emulation framework, supporting finer-scale climate impact assessments and informing local adaptation and mitigation strategies.

## 1 Introduction

Warming at regional scales is a critical aspect of climate change as humans and natural systems are affected by local temperatures and not the global mean. Warming patterns vary spatially with geographical features, with amplification in urbanized and deforested regions (IPCC, 2023), as well as in mountainous areas (Mountain Research Initiative EDW Working Group, 2015). These heterogeneous warming patterns necessitate finer-scale climate projections, often achieved using Regional Climate Models (RCMs) downscaling Global Climate Model (GCM) simulations (Soares et al., 2022; Careto et al., 2022; Jacob et al., 2020; Roberts et al., 2019). However, these high-resolution simulations are computationally expensive and the need to explore uncertainty within multiple models and scenario pathways significantly increases computational demands. In addition, representing internal climate variability requires large initial-condition ensembles (Eyring et al., 2024; Schneider et al., 2023; Giorgi, 2019; Hawkins and Sutton, 2009), further escalating the overall cost.

Recent advancements in climate emulators address these challenges by simplifying model complexity, thus enabling rapid climate projections under various scenario pathways. Among these, RCM emulators have gained attention as efficient surrogates for RCMs, providing insights into regional climate projections. While both RCM emulators and statistical downscaling methods aim to bridge the gap between coarse-resolution GCM outputs and fine-scale regional climate information, they differ



in their setup and objectives. RCM emulators are trained on GCM-RCM model chain simulations to reproduce RCM outputs, whereas statistical downscaling methods rely on empirical relationships derived from observational data to establish connec-

tions between large-scale and local-scale climate variables. Thus, RCM emulators and statistical downscaling methods can inform each other. Leveraging both deep neural networks (Doury et al., 2023, 2024; Van Der Meer et al., 2023; Baño-Medina et al., 2022; Passarella et al., 2022) and classical statistic approaches (Hobeichi et al., 2023; Boé and Mass, 2022), recent advancements in RCM emulators have achieved remarkable accuracy and efficiency. Moreover, efforts have been made to enhance interpretability of RCM emulators (González-Abad et al., 2023; Baño-Medina et al., 2022) and to evaluate transferability

across emission scenarios, initial-condition ensembles, and GCMs (Hernanz et al., 2024, 2022; Bano-Medina et al., 2023).

Despite these advancements, current RCM emulators focus primarily on deterministic prediction and do not fully account for the internal RCM variability, which is essential for useful regional climate projections. Recently, deep learning–based generative models have shown promising performance in probabilistic ensemble downscaling by enabling realistic sampling of internal variability (Lopez-Gomez et al., 2024; Watt and Mansfield, 2024; Ling et al., 2024; Bischoff and Deck, 2024; Tomasi

et al., 2024; Mardani et al., 2024; Wan et al., 2023). However, the inherent complexity of these models poses challenges for interpretability and requires significant computational resources during training and inference. Furthermore, these highly flexible models may fail to achieve optimal performance due to the currently limited sample size of GCM-RCM simulations that are available for training. To address these gaps, we present MESMER-RCM, a probabilistic ensemble RCM emulator designed for spatially-resolved annual 2m temperature projections. MESMER-RCM is a generative model that achieves both

interpretability and data-efficient learning. This study focuses on an RCM emulator tailored for the Europe (EURO-CORDEX, Gutowski et al. (2016)), with potential applicability to other regions. MESMER-RCM can be integrated with GCM emulators such as MESMER (Beusch et al., 2020b), providing a foundation for building an robust GCM–RCM emulator chain for future applications.

## 2 Data

MESMER-RCM training and validation is based on GCM-RCM model chain simulations from the EURO-CORDEX-CMIP5 experiment (Gutowski et al., 2016; Taylor et al., 2012). We use RCM simulations with a spatial resolutions of 0.44° and re-grid GCM data to a common resolution of 2.5°. The dataset aligns with the one used in CH2018 Technical Report (2018). This study focuses on annual mean 2 meter air temperature, for the period 1979 to 2099, and land grid points in the RCM domain. Each RCM simulation consists of 129 years of data and land 5929 grid points. The dataset contains 42 simulations (Table S1),

from five RCMs and nine GCMs under the Representative Concentration Pathways (RCP) 2.6, 4.5, and 8.5 scenario pathways (Meinshausen et al., 2011).

The simulation matrix for GCM-RCM combinations in EURO-CORDEX is sparse, with SMHI-RCA4 dominating the available model chains. Additionally, not all model chains provide all scenarios and there are only few which contributed initial-condition ensembles. These limitations and their implications for the methodology are discussed in detail in section 5.3. Despite

these limitations, the dataset provides a diverse set of GCM-RCM combinations under key scenario pathways, making it suit-





able to evaluate the performance of MESMER-RCM in emulating different GCM-RCM model chains and exploring various scenario pathways.

## 3 MESMER-RCM: A framework for regional climate model emulation

### 3.1 MESMER-RCM structure

MESMER-RCM explicitly models the annual mean temperature at an RCM grid point $i$ and time step $t$ as a linear combination of the RCM response to the driving GCM and a residual climate variability term:

$$T_{i,t}^{RCM} = f(T_{s,t}^{GCM}) + \eta_{i,t}^{RCM} \tag{1}$$

where $f(T_{s,t}^{GCM})$ denotes the deterministic response of the RCM at grid point $i$ to the GCM temperature at grid point $s$, and $\eta_{i,t}^{RCM}$ represents the residual variability. The deterministic response term captures the systematic relationship between GCM

and RCM simulations, while the residual variability term approximates the additional internal variability implied by the RCM as a stochastic process.

### 3.2 Deterministic response module

We use Lasso multiple linear regression (Lasso MLR) to predict the deterministic RCM temperature response of Eq. (1):

$$f(T_{s,t}^{GCM}) = a_i^{int} + \sum_{s \in N_k(i)} a_{i,s} T_{s,t}^{GCM} \tag{2}$$

The predictors are the 2m temperature at the $k$ GCM grid points closest to the RCM grid point $i$ (i.e., $s \in N_k(i)$). Here, $a_{i,s}$ represents the scaling coefficients that determine the influence of GCM grid points on RCM grid point $i$, while $a_i^{int}$ captures the local intercept. The scaling coefficients $a_{i,s}$ for corresponding GCM predictors are optimized by minimizing a lasso loss that combines the least squares term and an L1 normalization term, $\lambda \cdot \sum_{s=1}^{k} |a_{i,s}|$ (James et al., 2013). The L1 regularization adaptively selects the most relevant GCM grid points for predicting RCM temperature while shrinking the scaling coefficients of less relevant contributions towards zero. The Lasso coefficient $\lambda$ is determined through 5-fold cross-validation based on

mean absolute error (James et al., 2013). In this study, we select the nearest nine GCM grid points of RCM grid point as predictors ($k = 9$), forming a $3 \times 3$ grid that captures local GCM information around each RCM grid point.

### 3.3 Residual variability module

The residual variability term, $\eta_{i,t}^{RCM,res}$, is assumed to follow a multivariate Gaussian distribution, $\eta_{i,t}^{RCM,res} \sim \mathcal{N}(0, \Sigma_{RCM}^{res})$,

where $\Sigma_{RCM}^{res}$ denotes the spatial covariance matrix across all considered RCM grid points, that is estimated from the residuals after subtracting the deterministic response. However, since each RCM simulation includes far more grid points than sample years, the sample covariance matrix of the residuals $\hat{\Sigma}_{RCM}^{res}$ becomes rank-deficient and needs to be regularized to improve the accuracy and stability of the estimate (Stein, 1956; Ledoit and Wolf, 2003; Pourahmadi, 2011). This is often achieved by





shrinking the sample covariance matrix $\hat{S}$ to a target covariance matrix $T$ that encapsulates prior knowledge on the dependence structure of the underlying process such that

$$S = \alpha \cdot T + (1 - \alpha) \cdot \hat{S} \tag{3}$$

where $\alpha$ is the shrinkage parameter. Typical shrinkage targets are diagonal matrices such as the unit matrix since they are full rank. However, these assume independence for non-diagonal entries of the covariance matrix, which is physically not consistent with climate variability that is strongly correlated in space.

To address this limitation, we propose to leverage the full ensemble of RCM simulations to construct a data-driven prior, $P$, for $T$. The prior $P$ is estimated by first concatenating the residuals of all RCM simulations, except those used for training and testing, and subsequent computations of an sample covariance matrix. Due to the increased sample size, $P$ is nearly full rank and incorporates knowledge on the covariance structure derived from the available RCM simulations. However, $P$ remains rank deficient and hence needs and is shrunken to $\Lambda = diag(\hat{\Sigma}_{\text{RCM}}^{\text{res}})$ which incorporates prior knowledge that sample variances typically have lower estimation errors than the covariances. The resulting shrinkage estimator can thus be written as

$$\Sigma_{\text{RCM}}^{\text{res}}(\alpha, \beta) = \alpha \cdot [(1 - \beta) \cdot P + \beta \cdot \Lambda] + (1 - \alpha) \cdot \hat{\Sigma}_{\text{RCM}}^{\text{res}} \tag{4}$$

where the coefficients $\alpha$ and $\beta$ are in [0,1] and are jointly optimized by maximizing the Gaussian log-likelihood in a grid search using 5-fold cross-validation.

## 4 Applying MESMER-RCM

### 4.1 Modular training and emulation workflow

MESMER-RCM follows a modular training approach, where the deterministic response and the residual variability module are learned separately. The default training setup uses a single GCM-RCM simulation pair to learn the parameters of the deterministic response module and the sample covariance matrix $\hat{\Sigma}_{\text{RCM}}^{\text{res}}$ in the residual variability module. In contrast, the prior $P$ is constructed by concatenating multiple RCM residuals. The pooled approach for $P$ ensures numerical stability, but may dilute the chain-specific characteristics of residual variability. This trade-off is further examined through targeted sensitivity experiments (section 4.3).

To generate an ensemble of $M$ RCM emulations, we first apply the deterministic response module (Eq. 2), replicate its output $M$ times, and then add spatially correlated innovations drawn from the residual variability module (Eq. 4) to each copy.

### 4.2 Validation of MESMER-RCM

This study focuses on validating MESMER-RCM within each GCM–RCM model chain. The default training setup is evaluated through cross-validation across ensemble members, where training and testing are systematically interchanged. Specifically, each ensemble member—spanning different scenario pathways and initial conditions—is used in turn for training to predict





another member, exhaustively covering all possible combinations. Thus, the number of validation experiments varies across GCM–RCM model chains due to differences in data availability.

Emulator performance is assessed using rank histogram, which are widely used to evaluate the reliability of ensemble forecasting systems Wilks (2019, 2006); jol (2012); Weigel et al. (2008). For each ensemble forecast (or here emulation) consisting of $M$ ensemble members, the verifying reference is combined with the ensemble members to form a vector of $M + 1$ values. The reference is ranked within this vector, where rank 1 the smallest and rank $M + 1$ to the largest value within the ensemble. If the reference is statistically indistinguishable from the ensemble predictions, then each rank is equally likely

to occur. In such cases, the rank histogram will be flat, apart from small fluctuations caused by sampling variability. Deviations from flatness in the rank histogram provide insights into miscalibration. In this study, the flatness of the rank histogram is tested at the grid-point level using the $\chi^2$ statistic Wilks (2019). To obtain an aggregated performance metric, we compute the $\chi^2$ test pass rate, defined as the percentage of grid points for which the null hypothesis of flatness is not rejected.

### 4.3    Sensitivity to Training Configurations

Given the sparsity of GCM–RCM simulations, it is crucial to evaluate how model performance varies with different training datasets. In this study, we conduct three sensitivity experiments that use different samples to train the deterministic response module, prior $P$ in the residual variability module, and both, respectively. (1) To assess the robustness of the deterministic response module (Eq. 2), we train it using concatenated GCM–RCM simulation pairs from all available ensemble members, except the pair currently used for testing. (2) The trade-off in the residual variability module (Eq. 4) is evaluated by constructing the prior $P$ from a reduced set of concatenated RCM residuals, selected for their higher physical consistency with the training

set. (3) Combined approach: Applying both (1) and (2) simultaneously. All sensitivity experiments follow the same cross-validation strategy as the default setup, in which each ensemble member is predicted using the emulator trained on other combinations of ensemble members, depending on the experiment. Performance differences are further evaluated using the $\chi^2$ test pass rate.

## 5    Results

### 5.1    Calibration of MESMER-RCM

The deterministic response module is calibrated for each grid-cell individually but working with the same Lasso coefficient ($\lambda = 0.005$) across all grid-cells and model chains. The common lasso coefficient is chosen based on a cross-validation exercise that the optimized Lasso coefficients exhibit spatial smoothness, with European land means stabilizing around 0.005 across all

simulations (Figure S1). A summary of the coefficients of the deterministic response module alongside some insights to their spatial variability is provided in the supporting information (Figure S2, S3). The sum of scaling coefficients ($\sum_{s \in N_k(i)} a_{i,s}$) shows higher values in mountainous regions and high-latitude areas, highlighting the capacity of deterministic response module to capture elevation-dependent warming and polar amplification, respectively. The scaling coefficients indicate that an RCM





grid point in the Swiss Alps warms faster than its northwest and southeast GCM counterparts. This pattern may be related to
the north/south foehn effect over the Alpine region (Chow et al., 2013).

The shrinkage coefficients $(\alpha, \beta)$ in residual variability module is calibrated for each model chain. The resulting log-likelihood field associated with $\alpha$ and $\beta$ are consistent across all model chains, where the optimum $\beta$ is around $10^{-2}$ for stabilizing $\hat{\Sigma}^{\mathrm{res}}_{\mathrm{RCM}}$ and $\alpha$ is around $10^{-1}$ for effectively borrowing prior information from $P$. An example of such log-likelihood field is provided in the supporting information (Figure S4)

**Figure 1.** Examples of RCM emulations. Panels a to d show the annual mean temperature anomaly field for 2030, including the GCM reference (a), RCM reference (b), and two example emulations (c-d). Panels e to g present time series of ensemble emulations at three mountainous RCM grid points, with GCM and RCM reference and the location shown in the inset. The range is computed using 1000 realizations of the emulator.

## 5.2 Example of ensemble RCM emulations


Figure 1 presents an example of the ensemble RCM emulations for an GCM-RCM model chain simulation (RCM: SMHI-RCA4, GCM: CanESM2). The emulated temperature fields closely resemble the RCM reference, providing finer-scale details, such as variations induced by topography. The time series further illustrates MESMER-RCM's ability to adjust the GCM trajectory to align with the RCM at the grid-point level, regardless of whether the driving GCM shows a smaller or lager trend.




Additionally, the ensemble range effectively captures the variability of the RCM reference, demonstrating MESMER-RCM's capability for sampling internal RCM variability.

Emulated Model Chain [GCM: CanESM2, RCM: SMHI-RCA4]
Train: RCP4.5, Test: RCP8.5; Ensemble member: r1i1p1

**Figure 2.** Evaluation of example model chain (RCM: SMHI-RCA4, GCM: CanESM2). The rank histogram based on all RCM grid points, where the red dash line indicates the ideal even distribution (a) and $\chi^2$ test for rank histogram flatness at the RCM grid-point level (b). The hatched area in the $\chi^2$ statistic map represents regions that fail the $\chi^2$ test (i.e., $\chi^2 > \chi^2_{crit}$).

## 5.3 Evaluation of the probabilistic RCM emulator

The validation of the example GCM-RCM model chain (RCM: SMHI-RCA4, GCM: CanESM2) is shown in Figure 2. The rank histogram (Figure 2a) shows spatially-aggregated results, where the flatness of the rank histogram indicates overall reliable

performance of the RCM emulator. The $\chi^2$ test, applied at the grid-point level, shows that 80% of RCM grid points exhibit significant rank uniformity, indicating generally reliable performance at the RCM grid-point level as well (Figure 2b).

Extending the validation to all GCM-RCM model chains (Figure 3), we observe fluctuations in the $\chi^2$ test pass rate across different model chains, with a relatively large spread within individual chains. This fluctuation can be largely explained by our conducted sensitivity experiments. In default setup, the example model chain SMHI-RCA4_CanESM2 is one of the better

performing ones, while e.g. KNMI-RACMO22E_EC-EARTH only has a pass rate of 40%. Sensitivity experiments show a clear improvement in emulator performance compared to the default training setup. Training the deterministic response module in multiple instead of a single simulation pairs (Figure 3b) improves the robustness of the scaling coefficients, which reduce bias in temperature trend projections. The resulting spread of $\chi^2$ pass rates across validation experiments is considerably lower. For example, in MPI-CSC-REMO2009_MPI-GCM-LR, the average $\chi^2$ test pass rate increases by $10\%$ compared to default

setup, while the spread decreases from $\pm20\%$ to $\pm5\%$. The next sensitivity experiment revisits the residual variability module, using a physically consistent subset of concatenated RCM residuals to construct the prior $P$ (Figure 3c). This improves the statistical representation of residual variability, leading to higher average $\chi^2$ test pass rates. For instance, the pass rate increases




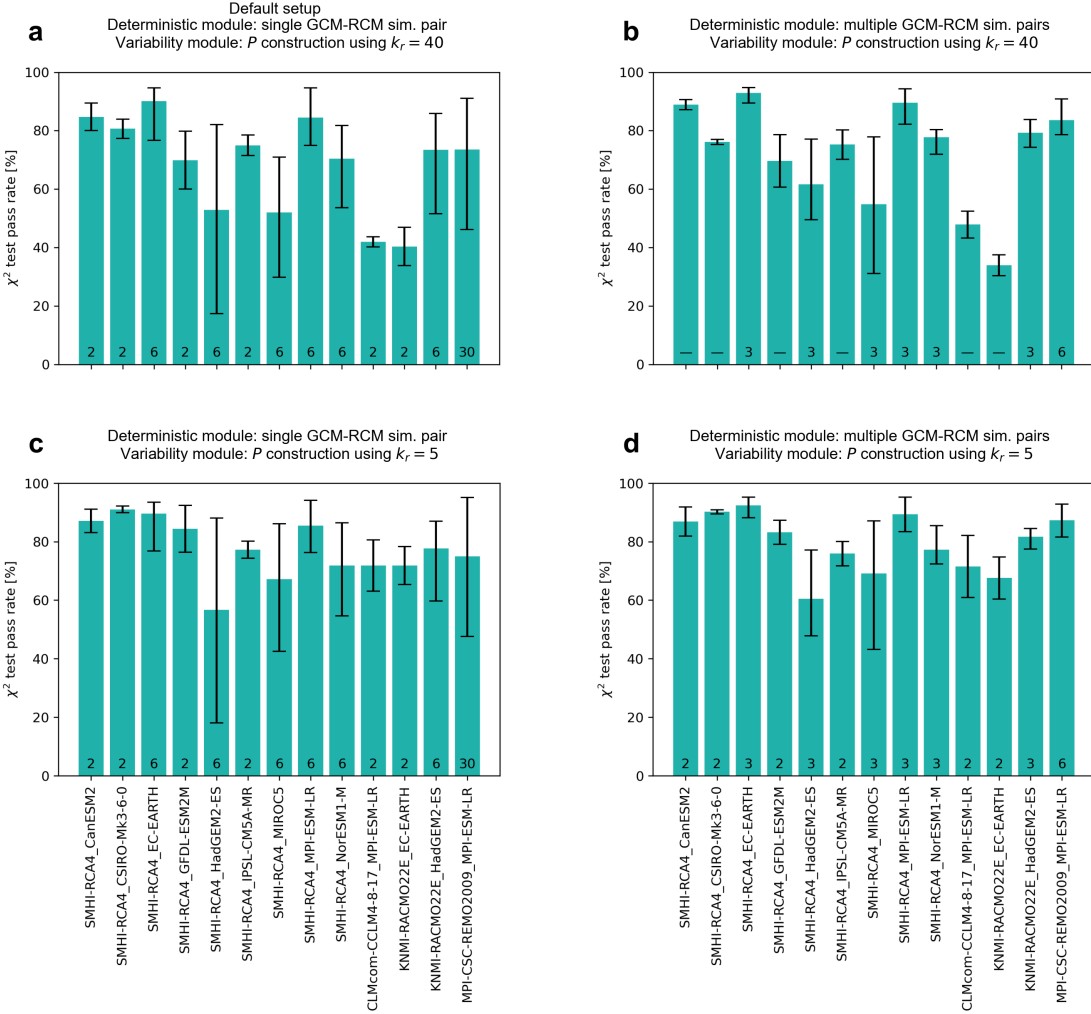

**Figure 3.** Comparison of training setups: $\chi^2$ test pass rates across different GCM-RCM model chains. The x-axis lists the model chains, formatted as RCM-name_GCM-name. The number above each chain indicating the total validation experiments conducted. A dash (—) denotes cases where the sensitivity experiment could not be applied to a model chain due to data availability. In such cases, the experiment has no effect on the model chain, and the performance shown corresponds to the default setup. The prior covariance matrix $P$ is constructed using residuals from the $k_r$ nearest RCM simulations, determined by the Euclidean distance between their $diag(\hat{\Sigma}_{\mathrm{RCM}}^{\mathrm{res}})$ and that of the training set. Here, $k_r = 40$ corresponds to the total number of available residuals of RCM simulations.

by 30% for KNMI-RACMO22E_HadGEM2-ES and CLMcom-CCLM4-8-17_MPI-ESM-LR. Due to limited data availability, the construction of prior $P$ faces a trade-off between physical consistency and numerical stability. In cases where stability is

compromised, the emulator may suffer from distortion due to ill-conditioned sampling. Nevertheless, this sensitivity experiment highlights the potential to alleviate this trade-off by identifying and incorporating diverse, physically consistent residuals into





the construction of $P$, thereby improving the sampling of residual variability. Finally, the combined approach (Figure 3d) shows a clear additive improvement in RCM emulation, highlighting the complementary strengths of training strategies indicated by sensitivity experiments. These results reveals that the observed inter-model performance variability primarily stems from
unevenly distributed GCM–RCM training data, and the emulator performance is generally robust when trained with well-prepared inputs.

## 6 Conclusions

This study develops a probabilistic RCM emulator, MESMER-RCM, that shows reliable performance in emulating annual 2m temperature projections. MESMER-RCM is capable of exploring regional-scale warming under various scenario pathways and
sampling internal RCM variability. Its modular structure enhances model interpretability and ensures computational efficiency, offering a simpler yet effective alternative to commonly used deep learning approaches. The sensitivity experiments conducted in this study highlight not only the influence of training data on emulator behavior, but also the importance of having sufficient GCM–RCM simulations to support the development of robust RCM emulators.

Future work on MESMER-RCM will leverage its ability to generate high-resolution ensemble projections from coarse-
resolution data. Specifically, MESMER-RCM can be further integrated with GCM emulators to form a GCM-RCM emulator chain, adding value to GCM emulators by enhancing their spatial resolution. For instance, MESMER generates spatially resolved annual mean temperature fields from arbitrary global mean temperature inputs. Coupling MESMER with MESMER-RCM enables regional climate projections across a wide range of GCM–RCM combinations and scenario pathways, thereby facilitating comprehensive uncertainty quantification. Such a framework holds strong potential for informing regional climate
risk assessments and supporting policy-relevant decision-making.

*Data availability.* CMIP5 data is accessible through https://wcrp-cmip.org/cmip-phases/cmip5/, and CORDEX simulations through https://cordex.org/data-access/cordex-cmip5-data/.

*Author contributions.* H.P., L.G., M.H. and S.I.S designed the study based on an initial idea from S.I.S. H.P. developed the methods with contributions from L.G. and M.H. H.P. wrote the manuscript with contributions from all authors.

*Competing interests.* The authors declare that they have no conflict of interest.



*Acknowledgements.* We acknowledge the World Climate Research Programme's Working Group on Regional Climate, and the Working Group on Coupled Modeling, former coordinating body of CORDEX and responsible panel for CMIP5. The authors gratefully acknowledge funding from the Joint Initiative SPEED2ZERO that received support from the ETH-Board under the Joint Initiatives scheme.



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
