# Peer review of "MESMER-RCM: A Probabilistic Climate Emulator for Regional Warming Projections"

_EGUsphere, 2025_

## Referee Comment (RC1)

This manuscript presents **MESMER-RCM**, a modular and computationally efficient probabilistic emulator for annual mean 2 m temperature at regional (EURO-CORDEX) scales. The approach is conceptually simple and interpretable: a Lasso-based deterministic mapping from nearby GCM grid points to each RCM grid point, combined with a data-driven, two-stage shrinkage estimator for the RCM residual covariance that enables sampling of spatially correlated internal variability. I think the illustrations in this manuscript is valuable, but there are still some issues need to be addressed before acceptance.

1. The authors need to clarify the MESMER method in detail, maybe in the supporting information. A schematic diagram is necessary.
2. The authors need to describe why you choose MESMER framework and its advantages compared with other generative AI models.
3. How to divide the training samples and testing samples in MESMER-RCM? And how many years for training and testing? In addition, please mark the time span of every dataset in Table S1.
4. Why the residual variability belongs to a multivariate Gaussian distribution? Is there any risk for underestimations under some extreme events? Please clarify this issue.
5. I think there should be another benchmark method to compare for further confirm the advantage of MESMER-RCM.
6. Could you provide more metrics on result evaluation? Such as CRPS.
7. Whether the residual covariance matrixes for different time ranges (e.g., 1979–2000, 2001–2050, 2051–2099) are various? If so, whether such variations are significant? Please clarify its effectiveness.

---

## Author Comment (AC1)

**Authors Response to Reviewer #1**

This manuscript presents MESMER-RCM, a modular and computationally efficient probabilistic emulator for annual mean 2 m temperature at regional (EURO-CORDEX) scales. The approach is conceptually simple and interpretable: a Lasso based deterministic mapping from nearby GCM grid points to each RCM grid point, combined with a data-driven, two-stage shrinkage estimator for the RCM residual covariance that enables sampling of spatially correlated internal variability. I think the illustrations in this manuscript is valuable, but there are still some issues need to be addressed before acceptance.

We thank Reviewer #1 for the accurate summary of this study and the constructive evaluation, which has helped us improve the manuscript.

1. The authors need to clarify the MESMER method in detail, maybe in the supporting information. A schematic diagram is necessary.

We thank Reviewer #1 question for guiding us to improve the scientific communication of this study. We improve the clarity of MESMER method, training and testing data preparation, and training procedure in sections 3, 4.1 and 4.2, respectively. We also designed a new schematic showcasing MESMER-RCM (Figure 1), which we believe to help convey MESMER-RCM methodology better.

2. The authors need to describe why you choose MESMER framework and its advantages compared with other generative AI models.

We thank Reviewer #1 for this question about MESMER-RCM framework and its significance. The MESMER-RCM framework is designed to be seamlessly coupled with existing global climate model output or the existing MESMER emulator, taking the MESMER emulation output (2m temperature field in GCM resolution) to generate a 2m-temperature field in RCM resolution at a comparably low computational cost. In line with MESMER, we designed MESMER-RCM such that model parameters to have a clear interpretation. For instance, the regression coefficient  $a_{i,s}$  represents the local scaling relationship between GCM and RCM that encapsulates the response signal of RCM temperature to GCM temperature. And the covariance matrix  $\Sigma^{res}_{RCM}$  describing the spatially correlated variability, which represents the intrinsic natural variability of the RCM.

3. How to divide the training samples and testing samples in MESMER-RCM? And how many years for training and testing? In addition, please mark the time span of every dataset in Table S1.

Regarding training samples and testing samples split, we correspondingly improved the description of training-testing experiment settings in section 4.1.

By default, 129 years are used for training and another 129 years for testing. However, we also perform sensitivity experiments on training sample size in section 4.4, where we use several ensembles members for training. For the sensitivity experiments we use  $n_{train}$  years for training,

where  $n_{train}$  is number of ensemble members. We keep single simulation pair for testing throughout this paper, hence the year for testing is 129 years. The time span for all GCM simulations is from 1850 to 2100 (251 years) and 1971 to 2099 (129 years) for all RCM simulations. We have now added this information to the caption of Table S1.

4. Why the residual variability belongs to a multivariate Gaussian distribution? Is there any risk for underestimations under some extreme events? Please clarify this issue.

We note that it is a well established and common practice to parameterize annual mean temperature variability using a normal distribution (e.g. von Storch & Zwiers 1999; Wilks 2006) and prior efforts within the MESMER framework have confirmed this practice (Beusch et al. 2020, 2022). Empirically, we have newly performed empirical diagnostics using a Shapiro-Wilk test for normality and show the percentage of RCM grid points that pass the test (new Figure S1), and the results suggest that the Gaussian assumption is a reasonable approximation for our residuals. The residual of RCM simulations, after removing the forced response, is the internal variability implied by the RCM. Such variability in the climate system is typically spatially correlated. A particular advantage of the multivariate Gaussian distribution is that it is fully determined by its mean and covariance, which allows us to capture both the central tendency and the spatial-temporal correlation structure of the residuals in a concise way.

We acknowledge that the multivariate Gaussian distribution has limitations in capturing extreme values. However, for modeling annual mean temperature, we consider this effect to be minor compared to its ability to represent the mean property. Since extreme signals are largely smoothed in yearly mean data, using a multivariate Gaussian distribution is unlikely to pose a substantial risk of underestimating extremes. One of the MESMER's emulator extensions, MESMER-X (Yann et al, 2022), is capable of emulating extreme temperature by assuming GEV distribution, which potentially can be extended to regional scale in the future work.

5. I think there should be another benchmark method to compare for further confirm the advantage of MESMER-RCM.

We have carefully designed new benchmarks for the deterministic response module and the residual variability module, respectively. Details of the benchmark design are provided in Section 4.3, and the corresponding comparison is presented in Figure 5.

6. Could you provide more metrics on result evaluation? Such as CRPS

We thank the reviewer for the suggestion to include additional evaluation metrics such as CRPS. In our study, we used the log-likelihood as a complementary metric to the rank histogram for evaluating MESMER-RCM. The rank histogram assesses whether the ensemble spread effectively captures the RCM reference at each grid point, whereas the log-likelihood measures how well the Gaussian distribution emulated by MESMER-RCM fits the RCM reference. We note that the log-likelihood serves a similar purpose to CRPS as a generic measure of distributional accuracy.

However, since MESMER-RCM explicitly assumes a parametric Gaussian distribution for the annual mean temperature, the log-likelihood is a more natural and theoretically consistent choice for assessing the fidelity of this probabilistic model.

7. Whether the residual covariance matrixes for different time ranges (e.g., 1979–2000, 2001–2050, 2051–2099) are various? If so, whether such variations are significant? Please clarify its effectiveness.

We thank the reviewer for this insightful question. In the MESMER framework, the residual covariance matrix is estimated based on the residuals of the training data as a whole and is therefore not explicitly dependent on the time period. This approach increases the stability of the covariance estimate in a high-dimensional setting, where dividing the data into shorter subperiods (e.g., 1979–2000, 2001–2050, 2051–2099) would lead to unreliable estimates dominated by sampling uncertainty. We acknowledge that this design implies an assumption of stationarity in the residual variability, meaning that potential non-stationarities in the covariance structure are not explicitly resolved. However, the relatively stable model performance in terms of log-likelihood across yearly test samples suggests that such effects are minor for our application and do not compromise the robustness of the MESMER-RCM emulation.

**Reference**

Storch, Hans von, and Francis W. Swiers. *Statistical Analysis in Climate Research*. Cambridge university press, 1999.

Wilks, Daniel S. *Statistical Methods in the Atmospheric Sciences*. 2nd ed. International Geophysics Series, volume 91. Elsevier, 2006.

Beusch, Lea, Lukas Gudmundsson, and Sonia I. Seneviratne. "Emulating Earth System Model Temperatures with MESMER: From Global Mean Temperature Trajectories to Grid-Point-Level Realizations on Land." *Earth System Dynamics* 11, no. 1 (2020): 139–59. https://doi.org/10.5194/esd-11-139-2020.

Beusch, Lea, Zebedee Nicholls, Lukas Gudmundsson, Mathias Hauser, Malte Meinshausen, and Sonia I. Seneviratne. "From Emission Scenarios to Spatially Resolved Projections with a Chain of Computationally Efficient Emulators: Coupling of MAGICC (v7.5.1) and MESMER (v0.8.3)." *Geoscientific Model Development* 15, no. 5 (2022): 2085–103. <a href="https://doi.org/10.5194/gmd-15-2085-2022">https://doi.org/10.5194/gmd-15-2085-2022</a>.

Quilcaille, Y.; Gudmundsson, L.; Beusch, L.; Hauser, M.; Seneviratne, S. I. Showcasing MESMER-X: Spatially Resolved Emulation of Annual Maximum Temperatures of Earth System Models. *Geophysical Research Letters* 2022, *49* (17), e2022GL099012. https://doi.org/10.1029/2022GL099012.

---

## Author Comment (AC2)

**Authors Response to Reviewer #2**

In this study, the authors presented a generative AI model for the region climate simulation. This model generates the large ensembles and can well capture the intrinsic climate variability. The topic is very interesting. But there still are several questions that need to be addressed.

We are grateful to Reviewer #2 for the clear summary of our study and for the insightful comments, which have been very helpful in revising and improving the manuscript.

1. How was the MESMER-RCM model trained? Please provide more details about the training of the model. For example, how to divide the training and testing sets? How to set the model parameters?

To improve clarity, we have restructured the manuscript: Section 3 now provides the methodological framework, while Sections 4.1 and 4.2 give detailed accounts of the data preparation and training procedure, including the division of training and testing data sets as well as the calibration of the model parameters. Furthermore, we have added a new Figure 1 to present a comprehensive schematic of MESMER-RCM. We believe these additions can improve the transparency and accessibility of the MESMER-RCM methodology.

2. There are some parameters in the model. Are the results sensitive to the choices of the parameters? The detailed tests should be done.

We added additional sensitivity experiments to clarify this. Regarding the deterministic response module, we examined the sensitivity to the number of nearest GCM grid points used as predictors, k. As shown in the new Figure 5a–c, we compared k=1 and k=9. Using k=9 provides a good balance between interpretability and emulation quality: it captures the local GCM-to-RCM temperature response effectively, while avoiding the blocky artifacts that typically arise from the simple nearest-neighbor regression (k=1). For the residual variability module, we tested the sensitivity to the number of residual samples used for prior P construction,  $k_r$  (Figure 4). Due to limited data availability, the choice of  $k_r$  reflects a trade-off between physical consistency and numerical stability: while increasing  $k_r$  enhances physical consistency and improves emulation performance (as indicated by the rank histogram), it compromises the numerical stability of the prior and may introduce distortions due to ill-sampling.

3. The figure 1 shows the 2-m temperature in a region. Why there are some blank areas? Additionally, could you add the latitude and longitude in the figure? Because the readers may be not familiar with that region.

We thank reviewer #2 for this helpful comment. MESMER focuses on emulating regional land warming within the EURO-CORDEX domain (-25°E to 45°E, 26°N to 72°N), which is consistently applied throughout this study. The blank areas in Figure 1 correspond to ocean regions outside the land domain. We have added latitude and longitude information in Section 2 (Data description) for clarity.

4. In this study, only a simple example was displayed. To show the advantage of the model, more examples in different areas should be presented. Whether can this model be extended to other regions?

We thank reviewer #2 for this valuable question. At present, EURO-CORDEX is the only region that provides a sufficiently large ensemble of simulations to meet the requirements for emulator training. MESMER-RCM is designed to be applicable, in principle, to any region, provided that a sufficiently large set of GCM—RCM model-chain simulations is available to construct the prior and thereby ensure physical consistency and numerical stability. Due to current data limitations, extending the emulator beyond Europe is not yet feasible. However, the upcoming CMIP6-CORDEX initiative is expected to provide a broader set of simulations, which would enable robust applications of MESMER-RCM in other regions, such as East Asia and North America. We highlight this as a promising avenue for future work in the conclusions.